# WIKI ENTITY SUMMARIZATION BENCHMARK

## ABSTRACT

Entity summarization aims to compute concise summaries for entities in knowledge graphs. However, current datasets and benchmarks are often limited to only a few hundred entities and overlook knowledge graph structure. This is particularly evident in the scarcity of ground-truth summaries, with few labeled entities available for evaluation and training. We propose WIKES (Wiki Entity Summarization Benchmark), a large *benchmark* comprising of entities, their summaries, and their connections. Additionally, WIKES features a dataset *generator* to test entity summarization algorithms in different subgraphs of the knowledge graph. Importantly, our approach combines graph algorithms and NLP models, as well as different data sources such that WIKES does not require human annotation, rendering the approach cost-effective and generalizable to multiple domains. Finally, WIKES is scalable and capable of capturing the complexities of knowledge graphs in terms of topology and semantics. WIKES features existing *datasets* for comparison. Empirical studies of entity summarization methods confirm the usefulness of our benchmark. Data, code, and models are available at: `https://anonymous.4open.science/r/Wikes-2DDA/README.md`.

## 1 INTRODUCTION

*Knowledge Graphs* (KGs) are a valuable information representation: interconnected networks of entities and their relationships that enable machine reasoning to empower question answering Hu et al. (2018); Lan et al. (2019), recommender systems Wang et al. (2018), information retrieval Raviv et al. (2016). KGs may comprise millions of entities representing real-world objects, concepts, or events.

Yet, the size and complexity of these KGs progressively expand, rendering it increasingly challenging to convey the essential information about an entity in a concise and meaningful way Suchanek et al. (2007); Vrandečić & Krötzsch (2014). This is where entity summarization becomes relevant. *Entity summarization* (ES) Liu et al. (2021) is the process of generating a concise and informative summary that captures the most salient aspects of the entity, based on the information available in the KGs. In ES, the entity *description* refers to all the triples involving such an entity. For instance, Figure 1 illustrates a set of relationships surrounding the entity `Ellen Johnson Sirleaf` in a KG, along with a possible summary for this entity. Extensive descriptions can overwhelm users and exceed the capacity of typical user interfaces, making it challenging to identify the most relevant triples. Entity summarization addresses this issue by computing an optimal compact summary for an entity, selecting a size-constrained subset of triples Liu et al. (2021).

Despite advancements in entity summarization techniques Liu et al. (2021), their development and evaluation face significant limitations in current benchmarks and datasets Liu et al. (2020); Cheng et al. (2023). First, existing benchmarks comprise only a few hundred entities, limiting dataset size. Second, generating ground-truth summaries primarily relies on costly and time-consuming manual annotation, which can introduce bias based on the preferences and knowledge of a few annotators. Lastly, these benchmarks often overlook the rich information contained within the knowledge graph structure.

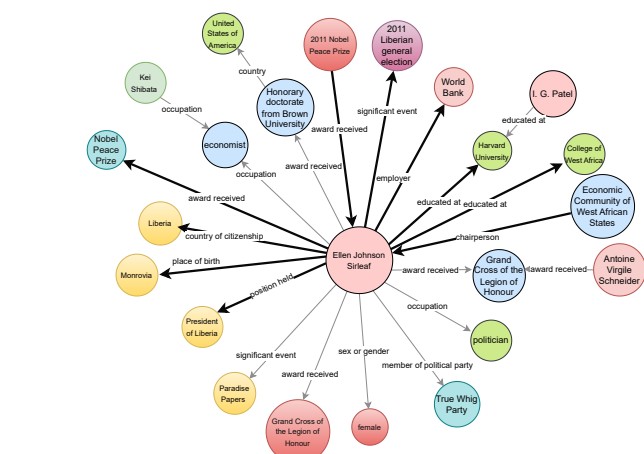

Figure 1: KG subgraph of entity `Ellen Johnson Sirleaf`: arrows depict the subgraph of relationships to other entities, and labels indicate their roles. Selecting the bold edges as entity summaries of the most relevant triples may reduce information overload while concisely describing the entity.

To address the above limitations, we propose:

- **Novel WIKES benchmark for ES** based on summaries and graphs from Wikidata and Wikipedia.
- **Subgraph extraction method** preserving the complexity of real-world KGs; subsampling using random walks and proportionally preserving node degrees, WIKES captures the structure of the entities up to the second-hop neighborhood, thereby ensuring that the connections in WIKES accurately reflect those in the source KG.
- **Comprehensive summaries for *any* entity in the KG**, ensuring that summaries are both relevant and contextually rich by deriving them directly from corresponding Wikipedia abstracts, minimizing human bias, as these abstracts are created and reviewed by several experts. In this manner, WIKES is scalable, enabling it to generate large benchmark resources efficiently with high-quality annotation.
- **Automatic entity summarization dataset generator** allows for the creation of arbitrarily large datasets, encompassing various domains of knowledge.

## 2 EXISTING DATASETS

Here, we review the existing datasets for entity summarization. Table 1 provides an overview and statistics of the current datasets in this field. FACES and INFO datasets have a higher density than the entities in the Entity Summarization Benchmark (ESBM). It is also clear that LMDB and FACES are not connected graphs, that challenge graph-based learning methods where the information cannot easily propagate in disconnected networks. Specifically, FACES consists of 12 connected components, which complicates the learning process for graph embedding methods by limiting the richness of information that can be leveraged from the graph.

We provide here a comprehensive description of each dataset or benchmark:

- **ESBM** Liu et al. (2020): The Entity Summarization Benchmark (ESBM) is the first benchmark to evaluate the performance of entity summarization methods. ESBM has three versions; v1.2 is the latest and most extensive version. This version comprises 175 entities, with 150 from DBpedia Lehmann et al. (2015) and 25 from LinkedMDB Hassanzadeh & Consens (2009). The summaries comprise triples selected by 30 "researchers and students" annotators. Each entity has exactly 6 summaries. Despite encompassing two

Table 1: Existing and WIKES datasets; number of entities $|\mathcal{V}|$, triples $|\mathcal{E}|$, ground-truth summaries, density $|\mathcal{E}|/\binom{|\mathcal{V}|}{2}$, number of components, sampling method, min/max node degree, and time to generate the dataset excluding preprocessing. RW refers to the random walk sampling method.

(a) Existing Datasets

| Metric | DBpedia (ESBM) | LMDB (ESBM) | FACES | INFO |
|---|---|---|---|---|
| $|\mathcal{V}|$ | 2721 | 1853 | 1379 | 1410 |
| $|\mathcal{E}|$ | 4436 | 2148 | 2152 | 2019 |
| Ground-truth | 125 | 50 | 50 | 100 |
| Density | 5e-4 | 6e-4 | 11e-4 | 10e-4 |
| Sampling | - | - | - | - |
| Components | 1 | 2 | 12 | 1 |
| Min Deg | 1 | 1 | 1 | 1 |
| Max Deg | 125 | 208 | 88 | 100 |

(c) WIKES **Small** Datasets

| Metric | WikiLitArt | WikiCinema | WikiPro | WikiProFem |
|---|---|---|---|---|
| $|\mathcal{V}|$ | 85346 | 70753 | 79825 | 79926 |
| $|\mathcal{E}|$ | 136950 | 126915 | 125912 | 123193 |
| Ground-truth | 494 | 493 | 493 | 468 |
| Density | 18e-6 | 18e-6 | 19e-6 | 19e-6 |
| Sampling | RW | RW | RW | RW |
| Components | 1 | 1 | 1 | 1 |
| Min Deg | 1 | 1 | 1 | 1 |
| Max Deg | 2172 | 3005 | 2060 | 3142 |
| Generation Time (s) | 91.9 | 118.0 | 126.1 | 177.6 |

(b) WIKES **Medium** Datasets

| Metric | WikiLitArt | WikiCinema | WikiPro | WikiProFem |
|---|---|---|---|---|
| $|\mathcal{V}|$ | 128061 | 101529 | 119305 | 122728 |
| $|\mathcal{E}|$ | 220263 | 196061 | 198663 | 196838 |
| Ground-truth | 494 | 493 | 493 | 468 |
| Density | 13e-6 | 19e-6 | 14e-6 | 13e-6 |
| Sampling | RW | RW | RW | RW |
| Components | 1 | 1 | 1 | 1 |
| Min Deg | 1 | 1 | 1 | 1 |
| Max Deg | 3726 | 5124 | 3445 | 5282 |
| Generation Time (s) | 155.4 | 196.4 | 208.2 | 301.7 |

(d) WIKES **Large** Datasets

| Metric | WikiLitArt | WikiCinema | WikiPro | WikiProFem |
|---|---|---|---|---|
| $|\mathcal{V}|$ | 239491 | 185098 | 230442 | 248012 |
| $|\mathcal{E}|$ | 466905 | 397546 | 412766 | 413895 |
| Ground-truth | 494 | 493 | 493 | 468 |
| Density | 8e-6 | 10e-6 | 8e-6 | 7e-6 |
| Sampling | RW | RW | RW | RW |
| Components | 1 | 1 | 1 | 1 |
| Min Deg | 1 | 1 | 1 | 1 |
| Max Deg | 8599 | 12189 | 7741 | 12939 |
| Generation Time (s) | 353.1 | 475.7 | 489.4 | 769.0 |

datasets, ESBM has several limitations. First, the entity sampling method is not explained. In particular, some triples in the neighborhood of the entity are missing in the datasets. Second, there are no connections among the entities in the neighborhood, nor any two-hop neighborhood. Third, the expertise and background of the annotators are not assessed nor disclosed. Due to the expensive annotation process, the dataset size is small.

- **FACES** Gunaratna et al. (2015) is a dataset from DBpedia (version 3.9) **?** and includes 50 randomly selected entities, each with at least 17 different types of relations. Similar to ESBM, the FACES ground-truth is also generated manually.
- INFO Cheng et al. (2023) features 100 randomly selected entities from 10 classes in DBpedia, including two sets of ground-truth summaries: REF-E and REF-W. REF-E summaries are crafted from triples by five experts with a 140-character limit, resembling Google search result snippets. In contrast, REF-W summaries are derived from one expert who reads Wikipedia abstracts and selects closely related neighboring entities. The number of ground-truth summaries per entity varies due to multiple evaluations by some experts, complicating the evaluation process. Additionally, the expertise of the annotators is not specified.

In contrast, our benchmark uses Wikidata to automatically map entities from Wikipedia to Wikidata. This automation allows us to efficiently generate summaries for any number of entities. Unlike previous work, we use the Wikipedia abstract as a summary instead of manual annotators. Each abstract is a collaboration of many users; as such, it should not introduce obvious biases. Additionally, with this process, we ensure high-quality and cost-effective summaries. Furthermore, we present the characteristics of our dataset in Table 1.The WIKES benchmark contains a significantly more entities and relations compared to existing datasets. Starting from approximately 500 target nodes, WIKES samples a connected graph, whereas existing datasets include *at most* 125 target nodes. Besides, LMDB and FACES are not connected graphs.

## 3 THE WIKES BENCHMARK

A *Knowledge Graph* $\mathcal{KG} = (\mathcal{V}, \mathcal{R}, \mathcal{T})$ is a directed multigraph consisting of entities $\mathcal{V} = \{v_1, \ldots, v_n\}$, relationships $\mathcal{R}$, and triples $\mathcal{T} \subseteq \mathcal{V} \times \mathcal{R} \times \mathcal{V}$. The set of edges $\mathcal{E} = \{(i, j) \mid v_i, v_j \in \mathcal{V} \wedge \exists r \in \mathcal{R} \text{ s.t. } (v_i, r, v_j) \in \mathcal{T}\}$ contains pairs of nodes connected by a relationship.

The *t-hop neighborhood* $\mathcal{N}_t(v_i)$ of node $v_i$ is the set of nodes reachable from $v_i$ within $t$ edges when ignoring edge directions.

A *summary* for an entity $v_i$ is a subset $\mathcal{S}(v_i) \subseteq \Delta_t(v_i)$ of triples from the $t$-description of $v_i$, where the *t-description* of an entity $v_i \in \mathcal{V}$ in a knowledge graph $\mathcal{KG}$ is the set $\Delta_t(v_i) = \{(s, p, o) \in \mathcal{T} \mid s \in \mathcal{N}_t(v_i) \vee o \in \mathcal{N}_t(v_i)\}$ of triples in which one of the entities is in the $t$-hop neighborhood of $v_i$.

**Entity summarization** Liu et al. (2021) for an entity $v_i \in \mathcal{V}$ in a knowledge graph $\mathcal{KG}$ aims to find a summary $\mathcal{S}(v_i)$ that maximizes a relevance score among all possible summaries for $v_i$, i.e.,

$$\underset{\substack{\mathcal{S}(v_i) \subseteq \Delta_t(v_i) \\ |\mathcal{S}(v_i)| = k}}{\arg\max} \ \text{score}(\mathcal{S}(v_i)), \tag{1}$$

The scoring functions differ among entity summarization methods, with some focusing on centrality and diversity of neighbors Cheng et al. (2011) and others employ PageRank-like scores Thalhammer et al. (2016).

## 3.1 Extracting Summaries from Wikidata using Wikipedia Abstracts

We extract summaries for each Wikidata item using Wikipedia abstracts and infoboxes. Each abstract is a joint effort of many users and experts, which ensures quality and accuracy. Leveraging Wikipedia, we avoid time-consuming manual annotation and enable the automatic generation of large-scale datasets.

**Wikidata** is a free and collaborative knowledge base that collects structured data to support Wikipedia and other Wikimedia projects. It includes descriptions and labels for entities. The descriptions offer in-depth details, while the labels serve as concise identifiers, facilitating efficient data retrieval and integration in subsequent steps. We load all Wikidata items XML dump files published on 2023/05/01[1] as entities $\mathcal{V}$ alongside their properties as relationships $\mathcal{R}$ into a graph database[2]. The result is a graph that connects all Wikidata items and statements. We include items if they (1) are not marked as redirects, (2) belong to the main Wikidata namespace, and (3) have an English label or description. Additionally, we load metadata for each Wikidata item and property, including labels and descriptions, into a relational database[3].

**Wikipedia** pages contain infoboxes, abstracts, page content, categories, references, and more. Links to other Wikipedia pages are referred to as mentions. We detect these mentions in the abstracts and infoboxes of Wikipedia pages to use them later for labeling the summaries in Wikidata. We extract and load all the content from the XML dump files of Wikipedia pages, published on 2023/05/01[4], into a relational database under the same conditions as Wikidata: the pages must be in English and not redirected.

**Summary annotation.** We annotate the summaries in Wikidata using the corresponding Wikipedia pages. For each Wikipedia page corresponding to a Wikidata entity, we iterate through all connected Wikidata items using Wikidata properties. If a connected Wikidata item is mentioned in the Wikipedia abstract and infobox, we annotate the Wikidata item with the corresponding Wikidata property as part of the summary.

Wikidata is a directed multigraph, which means that each entity (Wikidata item) can be connected to another entity via multiple relations (Wikidata properties). Yet, links in Wikipedia are not labeled; as such, we need to select one of the relations for the summary. To annotate the correct Wikidata property as part of the summary, we employ the DistilBERT model Sanh et al. (2019). DistilBERT is a fast and lightweight model with a reduced number of parameters compared to the original BERT model. This way, we can efficiently process large amounts of data while maintaining high-quality embeddings for accurate relation selection.

---

[1]https://dumps.wikimedia.org/wikidatawiki/
[2]https://neo4j.com
[3]https://www.postgresql.org/
[4]https://dumps.wikimedia.org/enwiki/

Concretely, we first embed the abstract of the Wikidata item for which we are generating summaries using DistilBERT. We then calculate the cosine similarity between the embedding of the abstract and the embeddings of each candidate relation. Finally, we add the relation with the highest cosine similarity to the abstract embedding to the summary. This approach ensures that the most relevant Wikidata property is selected for the summary based on its semantic similarity to the Wikipedia abstract.

## 3.2 CAPTURING THE GRAPH STRUCTURE

Here we introduce the WIKES generator algorithm. The main idea is to sample a connected graph that preserves the original graph structure. To this end, we employ random walks Pearson (1905). The random walk model is a straightforward yet effective method for preserving graph structure. While more recent techniques may yield superior results, we choose to use this widely accepted and fundamentally sound approach that exhibits good results even with $1\%$ sampled nodes (Figure 3).

A random walk is a stochastic process defined as a sequence of steps, where the direction and magnitude of each step are determined by the random variable $X_{t+1} = X_t + S_t$ where $X_t$ represents the position at time $t$, and $S_t$ is the step taken from position $X_t$.

The process is a Markov process, characterized by its memoryless property:

$$P(X_{t+1} = x | X_t = x_t, X_{t-1} = x_{t-1}, \ldots, X_0 = x_0) = P(X_{t+1} = x | X_t = x_t) \tag{2}$$

In adapting this concept to our work, we redefine the number of random walks assigned to nodes based on their degrees, ensuring the distribution remains proportional to real data. This is achieved through logarithmic transformation and normalization. The logarithmic transformation is applied to reduce the impact of high-degree nodes and also low-degree nodes, making it more manageable for the random walk. Given a graph with node degrees $\{d_1, d_2, \ldots, d_i\}$, the log-transformed degree for node $i$ is $L_i = \log(d_i)$. These values are then normalized:

$$N_i = \frac{L_i - \min(\{L\})}{\max(\{L\}) - \min(\{L\})} \tag{3}$$

where $N_i$ is the normalized logarithmic degree of node $i$. Finally, the number of random walks $R_i$ assigned to each node is:

$$R_i = \text{round} \left( \text{minRW} + N_i \times (\text{maxRW} - \text{minRW}) \right) \tag{4}$$

Here, minRW and maxRW are the user-defined minimum and maximum limits for random walks. This adaptation ensures that the random walks are proportional to the normalized logarithmic degree of each node, reflecting the true structure of the network. For a small dataset we set minRW $= 100$ and maxRW $= 300$; for a medium dataset minRW $= 150$ and maxRW $= 600$; for a large dataset, minRW $= 300$ and maxRW $= 1800$. This ensures that the random walks are tailored to both the scale and the complexity of the dataset. Importantly, our approach can be used to extract further subgraphs at the scale needed for benchmarking in a given scenario.

Moreover, the random walk sampling process requires a set of seed nodes as a starting point. In our case, the seed nodes represent the target entities we are interested in. The seed nodes can be any Wikidata Item Identifier, Wikipedia title, or Wikipedia ID of the Wikipedia pages. We collect the seed nodes on the condition that they have at least $k$ (default $k = 5$) common entities with the abstract section and the infobox in the Wikipedia pages. Therefore, this model is flexible, allowing you to choose any seed nodes from any domain as an input. In the datasets that we generated, we collect seed nodes from Laouenan et al. (2022). This paper has published information about individuals from various domains. The authors collected data from multiple Wikipedia editions and Wikidata, using deduplication and cross-verification techniques to compile a database of 1.6 million individuals with English Wikipedia pages. The seed nodes that we use include actor, athletic, football, journalist, painter, player, politician, singer, sport, writer, lawyer, film, composer, novelist, poet, and screenwriter. Using combinations of these seed nodes, we generate four sets of datasets, with each set having small, medium, and large versions. In Table 8 in Section A in the supplementary material, we present the seed nodes and their proportions for each dataset and their corresponding train-test-val splits.

### 3.3 WIKES GENERATOR

We discuss how WIKES is created, and how further benchmarks can be generated without the need for manual annotators. Algorithm 1 details the generator, which consists of the following steps.

**Step1:** Retrieve summaries of each seed node (explained in Section 3.1)

**Step2:** Expand the graph using the random walk method in Section 3.2. Set the random walk's length $n$ (default $n = 2$), which means it explores up to the $n$-hop neighborhood of each seed node. We choose $n = 2$ because extending beyond two hops risks introducing irrelevant entities, while our approach balances efficiency and accuracy. This ensures scalability and relevance for large datasets like Wikidata, complementing existing benchmarks Lissandrini et al. (2018).

**Step3:** Check if the graph is connected. If it is, done. If not, identify all disconnected components and sort them by size, from largest to smallest. In each iteration, connect smaller components to the largest component using $h$ connections. Utilize the shortest path method, selecting paths that are equal to or less than a minimum path length $l$. Continue connecting nodes from the smaller component to the larger one until $h$ nodes are connected. After each iteration, check graph connectivity again. If all components are connected to the largest component, the algorithm ends. Otherwise, re-sort components and increase $l$ by 1. Repeat until the graph is a single connected component.

---

**Algorithm 1** WIKES Generator

---

1: **Input:** Graph $G$, seed nodes $S$, random walk length $n$, minimum path length $l$
2: **Output:** A connected graph
3: **procedure** GENERATEGRAPH($G, S, n, l$)
4:     $summaries \leftarrow$ RETRIEVESUMMARIES($S$)
5:     $G \leftarrow$ RANDOMWALKEXPANSION($G, S, n$) mentioned in section 3.2
6:     $is\_connected \leftarrow$ CHECKCONNECTIVITY($G$)
7:     **while** not $is\_connected$ **do**
8:         $components \leftarrow$ FINDCOMPONENTS($G$)
9:         Sort $components$ by size in descending order
10:         $largest \leftarrow components[0]$
11:         **for** $comp$ in $components[1:]$ **do**
12:             Connect $comp$ to $largest$ using $h$ connections via shortest paths $\leq l$
13:             $G \leftarrow$ UPDATEGRAPH($G, comp, largest$)
14:             $is\_connected \leftarrow$ CHECKCONNECTIVITY($G$)
15:             **if** $is\_connected$ **then**
16:                 **break**
17:             **end if**
18:         **end for**
19:         $l \leftarrow l + 1$
20:     **end while**
21:     **return** $G$
22: **end procedure**

---

### 3.4 WIKES DATASETS

We generate three sizes for each of the four datasets, obtaining 12 datasets. We present their characteristics in Table 1 in section A. The number of entities in the small datasets ranges from approximately $70k$ to $85k$, and the number of relations ranges from around $120k$ to $135k$. In the medium datasets, the number of entities ranges from $100k$ to $130k$, and the number of relations ranges from $195k$ to $220k$. The number of entities in the large datasets ranges from approximately $185k$ to $250k$, and the number of relations ranges from around $397k$ to $470k$. The average runtime for generating small graphs is approximately 128 seconds; for medium-sized graphs, it is approximately 216 seconds; and for large graphs, it is approximately 512 seconds. We construct the train-test-validation split for each dataset with $70\%$ for training, $15\%$ for testing, and $15\%$ for validation. Detailed information about the run time, as well as the number of nodes and relations for these splits, is available on our GitHub repository. All graphs in each train-test-validation splits are connected.

## 4 EMPIRICAL EVALUATION

We study the quality of WIKES using the following metrics:

**F-Score.** Let $\mathcal{S}_m$ the summary obtained by a summarization method and $\mathcal{S}_h$ the ground-truth summary. We compare $\mathcal{S}_m$ with $\mathcal{S}_h$ using the F1-score based on precision $P$ and recall $R$:

$$\text{F1} = \frac{2 \cdot P \cdot R}{P + R}, \text{ where P} = \frac{|\mathcal{S}_m \cap \mathcal{S}_h|}{|\mathcal{S}_m|} \text{ and R} = \frac{|\mathcal{S}_m \cap \mathcal{S}_h|}{|\mathcal{S}_h|} \tag{5}$$

The F1 score lies within [0,1]. High F1 indicates that $\mathcal{S}_m$ is closer to the ground-truth $\mathcal{S}_h$.

**Mean Average Precision (MAP).** This metric is particularly suitable for evaluating ranking tasks because it takes into account the order of the predicted triples. MAP calculates precision at each position $i$ in the predicted summary and averages these values over all relevant summary triples. It reflects both the relevance and the ranking quality of the predicted summaries. MAP, unlike F1-score, does not depend on a specific value of $k$. This makes it a robust metric for assessing how well a summarization method ranks the relevant triples.

$$\text{MAP} = \frac{1}{N} \sum_{n=1}^{N} \frac{\sum_{i=1}^{|\mathcal{S}_m^{(n)}|} \begin{cases} \text{Precision@}i(\mathcal{S}_h^{(n)}) & \text{if Rel}(n,i) \\ 0 & \text{otherwise} \end{cases}}{|\mathcal{S}_h^{(n)}|} \tag{6}$$

where $N$ is the total number of entities, $\mathcal{S}_h^{(n)}$ is the set of ground-truth summary triples for a particular entity $v_n$, $\mathcal{S}_m^{(n)}$ is the set of predicted summary triples for the entity $v_n$, Precision@$i$ is the precision at the $i$-th position in the predicted summary, and $\text{Rel}(n,i)$ indicates whether the $i$-th predicted triple for entity $v_n$ is relevant (i.e., it belongs to $\mathcal{S}_h^{(n)}$). MAP scores are in the range [0,1], where a higher MAP indicates better performance in terms of correctly predicting relevant summary triples. To account for the varying lengths of the ground-truth summaries in real-world data, we also calculate MAP and F-score (which we refer to as dynamic MAP and dynamic F-score) by setting the length of the generated summary ($|\mathcal{S}_m|$) equal to the length of the corresponding ground-truth summary ($|\mathcal{S}_h|$).

We analyze our dataset and compare it with the ESBM benchmark using statistical measures such as frequency and inverse frequency of entities and relations. We calculate the F-score and MAP score for the top-5 and top-10 of both the ESBM dataset and our WikiProFem. We choose top-5 and top-10 because we only have ground-truth summaries for top-5 and top-10 in the ESBM dataset. The F-score and MAP results for ESBM are presented in Figure 2. The statistics show that for DBpedia, the F-score using inverse relation frequency outperforms the random baseline by 0.15 for top-5 and by 0.34 for top-10. Furthermore, when using inverse entity frequency, DBpedia achieves an even higher F-score, surpassing the random baseline by 0.07 for top-5 and by 0.15 for top-10. For LMDB, we observe a similar trend when using inverse frequency. The F-score surpasses the random baseline by 0.10 for top-5 and by approximately 0.15 for top-10. Additionally, when employing entity frequency, LMDB achieves an F-score that is around 0.17 higher than the baseline for top-5 and 0.07 higher for top-10. The results demonstrate that ESBM exhibits a strong bias towards entity, reverse entity, and relation frequency. For Map score, we are exactly observing the same behavior for ESBM. We believe that the bias comes from the fact that the datasets are small, their second-hop neighborhood is not considered, and the relations between their first-hop neighbors are not considered. On the other hand, Figure 3 shows the F-score for top-5, top-10 and dynamic F-score on WIKES. Since the length of summaries varies with the abstract, we calculate the F-score for each seed node based on its summary length. Results show that WIKES F-score is close to random for different statistics, thus rejecting the hypothesis of obvious biases. We observe a minor bias towards node frequency in small datasets. Yet, as we increase the size of the dataset, this bias disappears. We observe a similar behavior with MAP in Figure 11 (in appendix). Furthermore, we use *the entire* Wikidata to measure the F-score for our seed nodes. Thus, importantly, we observe that our

dataset's F-score trend is comparable to that of the entire data, especially our large dataset. We also extracted the first-hop neighborhood of all our seed nodes and observed a small bias in the F-score top-5 and dynamic F-score. We conclude that adding the two-hop neighborhood makes the sample follow the graph distribution. Thus, WIKES is an unbiased benchmark that retains the source KG distribution.

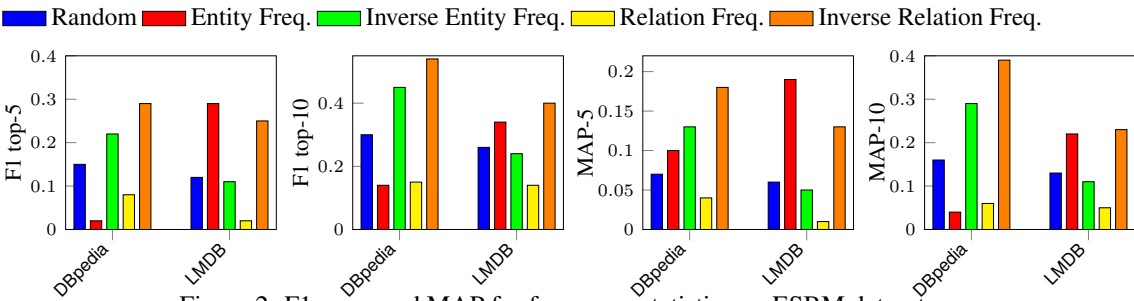

Figure 2: F1 score and MAP for frequency statistics on ESBM datasets.

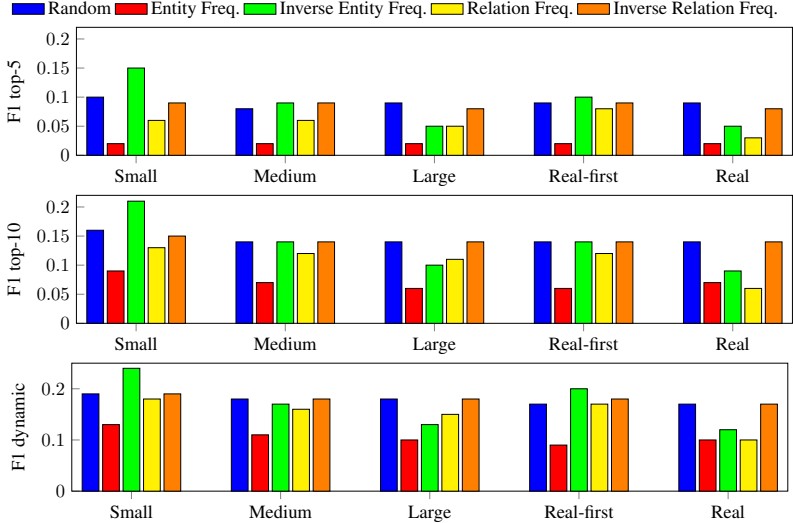

Figure 3: F1 for frequency statistics on WikiProFem.

We evaluate the performance of different entity summarization methods on our benchmark, and provide all implementations in the WIKES GitHub repository.

- **PageRank** Ma et al. (2008) is an unsupervised method that ranks nodes in a graph based on the structure of incoming links, with the idea that more important nodes are likely to receive more links from other nodes.

- **RELIN** Cheng et al. (2011) is another unsupervised approach, a weighted PageRank algorithm that evaluates the relevance of triples within a graph structure. We have re-implemented this model according to the specifications in the referenced paper. On our smaller dataset version, RELIN takes approximately 6 hours to compute all summaries.

- **LinkSum** Thalhammer et al. (2016) , also an unsupervised approach, is a two-step, relevance-centric method that combines PageRank with an adaptation of the Backlink algorithm to identify relevant connected entities. We have re-implemented it according to the paper. The LinkSum method initially takes 10 hours to compute the backlinks for each node in the small version of our dataset. By parallelizing the implementation, we

reduced this to one hour. Additionally, the Backlink algorithm itself initially takes 100 minutes, but with parallelization, this was reduced to 10 minutes for the small version of our dataset.

- **GATES** Firmansyah et al. (2021) is a recent *supervised* approach that integrates graph structure using Graph Attention Networks with knowledge graphs and text embeddings. We run GATES using the best performing hyperparameters of the original paper. GATES takes 20 minutes to run on the our small datasets.

We evaluate the methods on the smallest WIKES dataset due to their inefficiency. Table 2 shows that LinkSum, generally outperforms other models. Interestingly, GATES, despite being supervised, achieves lower accuracy compared to LinkSum. The deficiency in GATES may be due to its reliance on the frequency of nodes and relations, which are used as weights. As mentioned earlier, this frequency bias is present in ESBM but not in WIKES. These results highlight the significance of graph structure in summarizing entities within real-world knowledge graphs like WIKES, emphasizing the advantages of graph-based methods.

**Efficiency concerns.** The evaluation we conducted on various entity summarization models reveals significant efficiency issues with many recent baselines. For example, BAFREC Kroll et al. (2018), a model highlighted in a recent survey on unsupervised entity summarization, was unable to process a graph with 13 000 nodes — which is 5× smaller than our smallest dataset — even after running for two days. Similarly, MPSUM Wei et al. (2020) did not finish after 15 days on the same graph. Additionally, models like INFO Cheng et al. (2023), which depend on unavailable external resources, were excluded from our evaluation. These results highlight the need for more scalable approaches that can efficiently handle large knowledge graphs without sacrificing performance or accuracy.

| Model | Dataset | topK = 5 | | topK = 10 | | Dynamic | |
|---|---|---|---|---|---|---|---|
| | | F-Score | MAP | F-Score | MAP | F-Score | MAP |
| PageRank | WikiLitArt | 0.024 | 0.01 | 0.081 | 0.02 | 0.175 | 0.046 |
| | WikiCinema | 0.003 | 0.001 | 0.041 | 0.005 | 0.146 | 0.028 |
| | WikiPro | 0.060 | 0.02 | 0.169 | 0.049 | 0.288 | 0.109 |
| | WikiProFem | 0.032 | 0.01 | 0.093 | 0.024 | 0.145 | 0.036 |
| RELIN | WikiLitArt | 0.093 | 0.035 | 0.148 | 0.054 | 0.208 | 0.080 |
| | WikiCinema | 0.071 | 0.023 | 0.127 | 0.038 | 0.209 | 0.068 |
| | WikiPro | 0.125 | 0.053 | 0.200 | 0.086 | 0.273 | 0.127 |
| | WikiProFem | 0.111 | 0.050 | 0.179 | 0.081 | 0.219 | 0.095 |
| LinkSum | WikiLitArt | 0.184 | 0.080 | 0.239 | 0.109 | 0.225 | 0.127 |
| | WikiCinema | 0.119 | 0.048 | 0.152 | 0.060 | 0.135 | 0.068 |
| | WikiPro | 0.249 | 0.127 | 0.347 | 0.190 | 0.350 | 0.242 |
| | WikiProFem | 0.195 | 0.097 | 0.236 | 0.127 | 0.213 | 0.136 |
| GATES | WikiLitArt | 0.110 | 0.052 | 0.167 | 0.087 | 0.236 | 0.090 |
| | WikiCinema | 0.085 | 0.036 | 0.131 | 0.051 | 0.231 | 0.082 |
| | WikiPro | 0.149 | 0.074 | 0.225 | 0.118 | 0.313 | 0.149 |
| | WikiProFem | 0.128 | 0.062 | 0.227 | 0.097 | 0.243 | 0.114 |

Table 2: Performance comparison of entity summarization models on the small version of WIKES. The models are evaluated with different topK values (5 and 10) and a dynamic setting.

## 5 CONCLUSION

We introduce WIKES (Wiki Entity Summarization Benchmark), a benchmark for KG entity summarization which provides a scalable dataset generator that eschews the need for costly human annotation. WIKES uses Wikipedia abstracts for automatic summary generation, ensuring contextually rich and unbiased summaries. It preserves the complexity and integrity of real-world KGs through a random walk sampling method that captures the structure of entities down to their second-hop neighborhoods. Empirical evaluations demonstrate that WIKES provides high-quality large-scale datasets for entity summarization tasks, and that it captures the complexities of knowledge graphs in terms of topology, making it a valuable resource for evaluating and improving entity summarization algorithms.

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
