# A  APPENDIX

## A.1  ADDITIONAL WIKES DETAILS

- **Dataset and Metadata:** The dataset is available at `https://anonymous.4open.science/r/Wikes-2DDA`. We generate four datasets in three sizes: small, medium, and large. Each size has an entire graph that includes all seed nodes and train-test-validation splits. In Table 8, we provide information on the proportion of seed nodes in each of the datasets. Moreover, Table 1 provides detailed information such as the number of entities $|\mathcal{V}|$, triples $|\mathcal{E}|$, ground-truth summaries, density, graph connectivity, number of components, sampling method used to select the entities and the subgraph, minimum and maximum node degree, and running time for each of the datasets. Moreover, metadata is also in the same GitHub repository.

- **Dataset Formats:** We generate our dataset in CSV format. The entity files are formatted according to Table 3. We also provide files containing target entities and their categories, as detailed in Table 7. The predicate files, described in Table 5, contain predicates along with their corresponding labels and descriptions. The triple file, presented in Table 6, includes the subject, object, and predicate IDs of the nodes (Wikidata items) and edges (Wikidata predicates). The ground-truth file, shown in Table 7, contains the subject, object, and predicate. Moreover, we provide the graph version of our dataset in GraphML and PKL formats in our release.

- **Preprocessing URL:** You can find our preprocessing code for cleaning and preparing Wikipedia and Wikidata at the following link: `https://anonymous.4open.science/r/Wikes-2DDA/wiki-entity-summarization-preprocessor/README.md`.

- **Authors responsibility statement and License:** The authors are held responsible for copyright infringement, but assume no responsibility or liability for any misuse of the data. This project is licensed under the CC BY 4.0 License. See here `https://anonymous.4open.science/r/Wikes-2DDA/LICENSE`

- **WIKES Generator Code:** The code for running the WIKES generator is available in the GitHub repository at `https://anonymous.4open.science/r/Wikes-2DDA/README.md`. The code allows to generate the same datasets as those provided in the paper or to create your own custom datasets.

- **Maintenance and Long Term Preservation** The authors of WIKES are dedicated to the ongoing maintenance and preservation of this dataset. This includes tracking and resolving issues identified by the community post-release. We will closely monitor user feedback through the GitHub issue tracker. The data is hosted on GitHub, ensuring reliable and stable storage.

- **Intended users:** The intended users are NLP and knowledge graph researchers who wish to generate summaries using the textual information of the entities (nodes) in knowledge graphs. The suitable use case for this dataset is evaluating entity summarization models to determine their ability to detect summaries accurately.

| Field | Description | Datatype |
|---|---|---|
| id | Incremental integer starting by zero | int |
| entity | Wikidata qid, e.g. 'Q76' | string |
| wikidata_label | Wikidata label (nullable) | string |
| wikidata_desc | Wikidata description (nullable) | string |
| wikipedia_title | Wikipedia title (nullable) | string |
| wikipedia_id | Wikipedia page id (nullable) | long |

Table 3: `{variant}-{size}-{dataset_type}-entities.csv` file contains entities. An entity is a Wikidata item (node) in our dataset. variant_index refers to the dataset id (detailed information is in our Github).

| Field | Description | Datatype |
|---|---|---|
| entity | id key in Table 3 | int |
| category | category | string |

Table 4: {variant}-{size}-{dataset_type}-root-entities.csv contains root entities. A root entity is a seed node described previously. variant_index refers to the dataset id (detailed infomation is in our Github).

| Field | Description | Datatype |
|---|---|---|
| id | Incremental integer starting by zero | int |
| predicate | Wikidata Property id, e.g. 'P121' | string |
| predicate_label | Wikidata Property label (nullable) | string |
| predicate_desc | Wikidata Property description (nullable) | string |

Table 5: {variant}-{size}-{dataset_type}-predicates.csv contains predicates. A predicate is a Wikidata property or a describing a connection. variant_index refers to the dataset id (detailed information is in our Github).

| Dataset | Seed Nodes Categories |
|---|---|
| WikiLitArt | **Entire graph:** actor=150, composer=35, film=41, novelist=24, painter=59, poet=39, screenwriter=17, singer=72, writer=57 |
| | **Train:** actor=105, composer=24, film=29, novelist=17, painter=42, poet=27, screenwriter=12, singer=50, writer=40 |
| | **Val:** actor=23, composer=5, film=6, novelist=4, painter=9, poet=6, screenwriter=2, singer=11, writer=8 |
| | **Test:** actor=22, composer=6, film=6, novelist=3, painter=8, poet=6, screenwriter=3, singer=11, writer=9 |
| WikiCinema | **Entire graph:** actor=405, film=88 |
| | **Train:** actor=284, film=61 |
| | **Val:** actor=59, film=14 |
| | **Test:** actor=62, film=13 |
| WikiPro | **Entire graph:** actor=58, football=156, journalist=14, lawyer=16, painter=23, player=25, politician=125, singer=27, sport=21, writer=28 |
| | **Train:** actor=41, football=109, journalist=10, lawyer=11, painter=16, player=17, politician=87, singer=19, sport=15, writer=20 |
| | **Val:** actor=9, football=23, journalist=2, lawyer=3, painter=3, player=4, politician=19, singer=4, sport=3, writer=4 |
| | **Test:** actor=8, football=24, journalist=2, lawyer=2, painter=4, player=4, politician=19, singer=4, sport=3, writer=4 |
| WikiProFem | **Entire graph:** actor=141, athletic=25, football=24, journalist=16, painter=16, player=32, politician=81, singer=69, sport=18, writer=46 |
| | **Train:** actor=98, athletic=18, football=17, journalist=9, painter=13, player=22, politician=57, singer=48, sport=14, writer=34 |
| | **Val:** actor=21, athletic=4, football=3, journalist=4, painter=1, player=5, politician=13, singer=11, sport=1, writer=5 |
| | **Test:** actor=22, athletic=3, football=4, journalist=3, painter=2, player=5, politician=11, singer=10, sport=3, writer=7 |

Table 8: Seed nodes categories for each dataset. "Entire graph" refers to using the seed nodes and generating the data without train-test-val splits. In train-test-val, each of the datasets is a single weakly connected graph.

| Field | Description | Datatype |
|---|---|---|
| subject | id key in Table 3 | int |
| predicate | id key in Table 5 | int |
| object | id key in Table 3 | int |

Table 6: `{variant}-{size}-{dataset_type}-triples.csv` contains triples. A triple is an edge between two entities with a predicate. variant_index refers to the dataset id (detailed information is in our Github).

| Field | Description | Datatype |
|---|---|---|
| root_entity | entity in Table 3 | int |
| subject | id key in Table 3 | int |
| predicate | id key in Table 5 | int |
| object | id key in Table 3 | int |

Table 7: `{variant}-{size}-{dataset_type}-ground-truths.csv` contains ground-truth triples. A ground-truth triple is an edge marked as a summary for a root entity.

## A.2 PARAMETERS FOR RUNNING THE WIKES GENERATOR

Table 9 shows the parameters required for running the WIKES Generator. The table provides a description of the parameters and their default values, where applicable. A detailed explanation of how to run the generator can be found in our GitHub repository.

| Parameter | Description | Default Value |
|---|---|---|
| min_valid_summary_edges | Minimum number of valid summaries for a seed node | 5 |
| random_walk_depth_len | Depth length of random walks (number of nodes in each random walk) | 3 |
| bridges_number | Number of connecting path bridges between components | 5 |
| max_threads | Maximum number of threads | 4 |
| min_random_walk_number | Minimum number of random walks for each seed node, explained | 100 for small, 150 for medium, and 300 for large |
| max_random_walk_number | Maximum number of random walks for each seed node | 300 for small, 600 for medium, and 1800 for large |

Table 9: Parameters for Running WIKES Generator

## A.3 TECHNOLOGIES

Table 10 presents the versions of the technologies and configurations that we use in this work.

## A.4 EXPERIMENTS

We include the experiments in Section 4 for all of our datasets below.

Table 10: Technology and Configuration Details for Daatset Generations

(a) Technologies Used: Software Versions and Data Sources

| Technology | Version/Details |
|---|---|
| Java | Version 21 |
| Spring Boot | Version 3 |
| Docker | Version 24.0.8 |
| Python | Version 3.10 |
| PostgreSQL | Version 16.3 |
| Neo4j | Version 5.20.0-community |
| Wikipedia XML Article Dump Files | Published by Wikimedia on 2023/05/01 |
| Wikidata XML Article Dump Files | Published by Wikimedia on 2023/05/01 |

(b) Hardware- Spec: Specifications of the AWS EC2 Instance (r5a.4xlarge)

| Specification | Details |
|---|---|
| vCPU | 16 (AMD EPYC 7571, 16 MiB cache, 2.5 GHz) |
| Memory | 128 GB (DDR4, 2667 MT/s) |
| Storage | 500 GB (EBS, 2880 Max Bandwidth) |

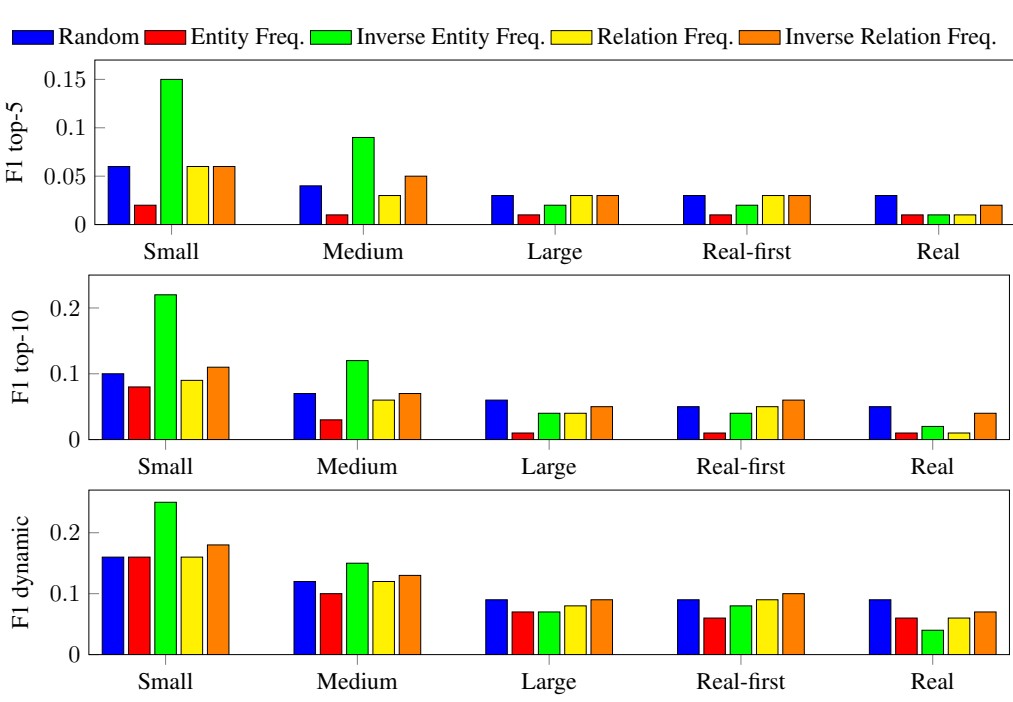

Figure 4: F1 for frequency statistics on WikiLitArt.

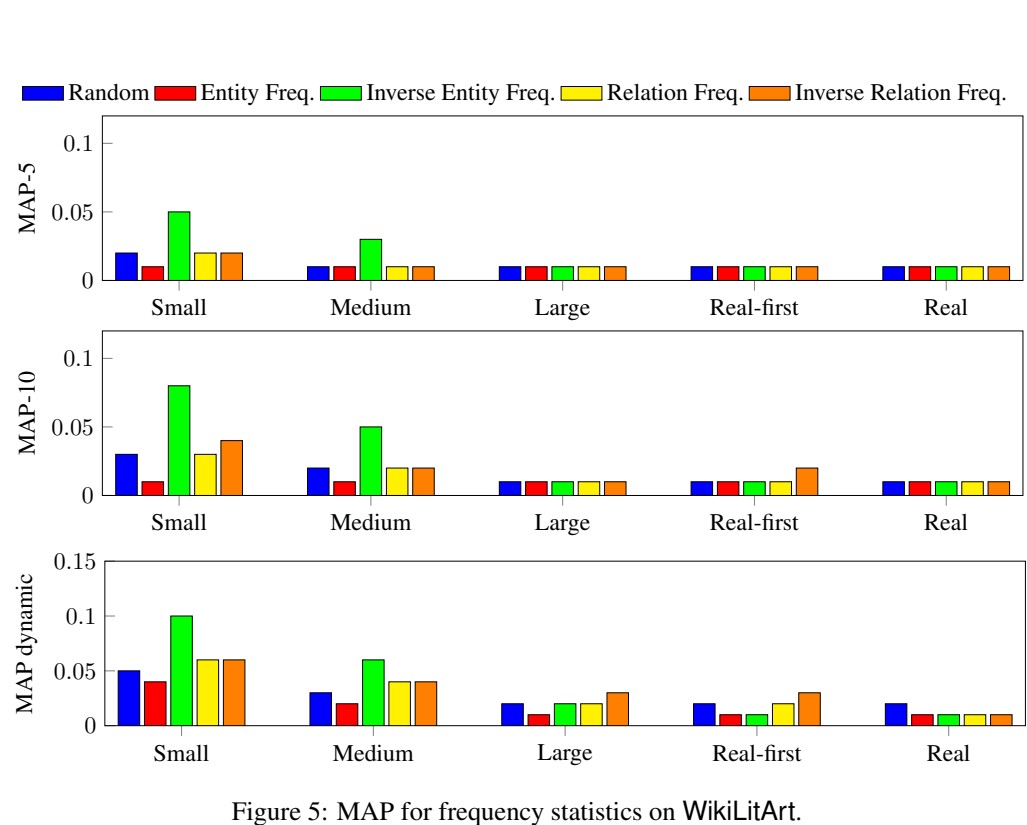

Figure 5: MAP for frequency statistics on WikiLitArt.

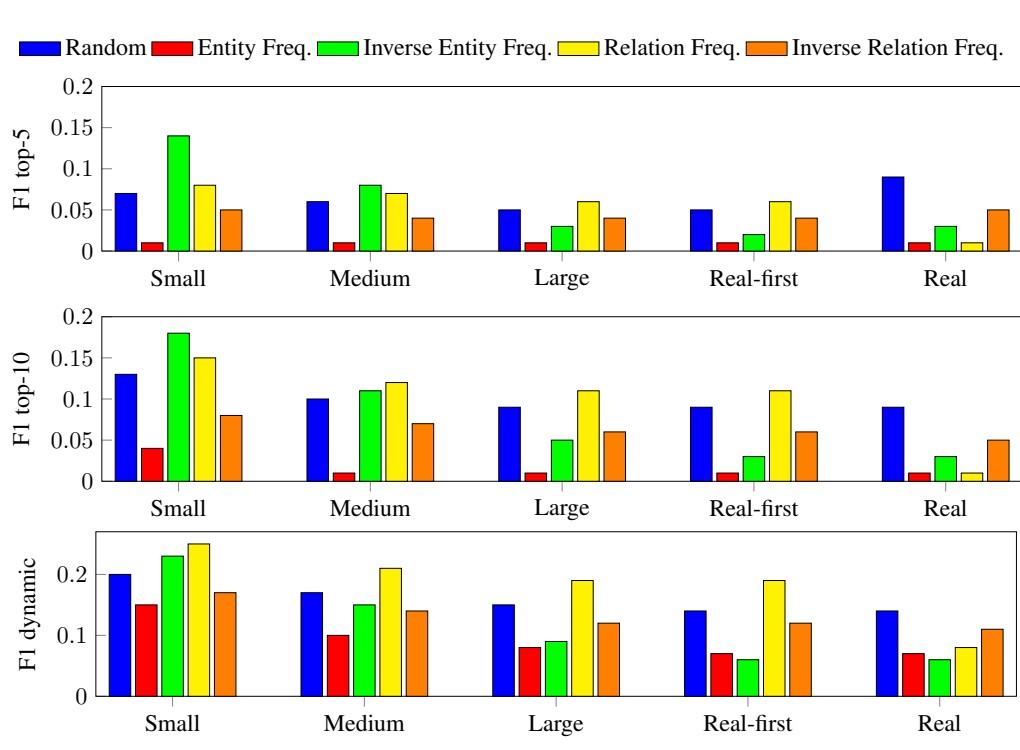

Figure 6: F1 for frequency statistics on WikiCinema.

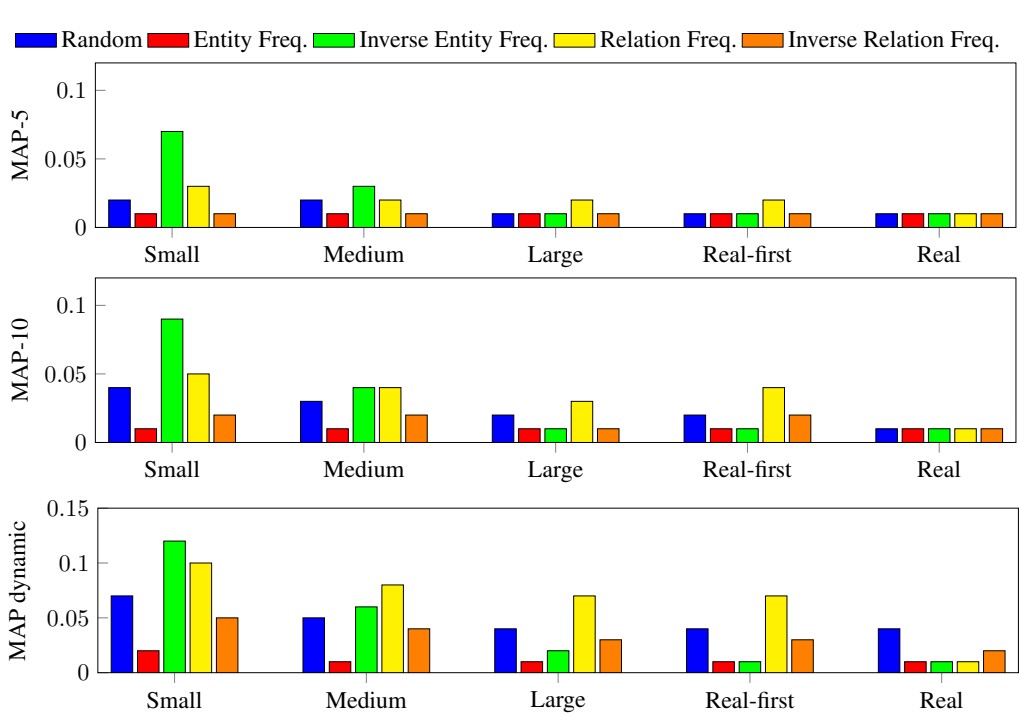

Figure 7: MAP for frequency statistics on WikiCinema.

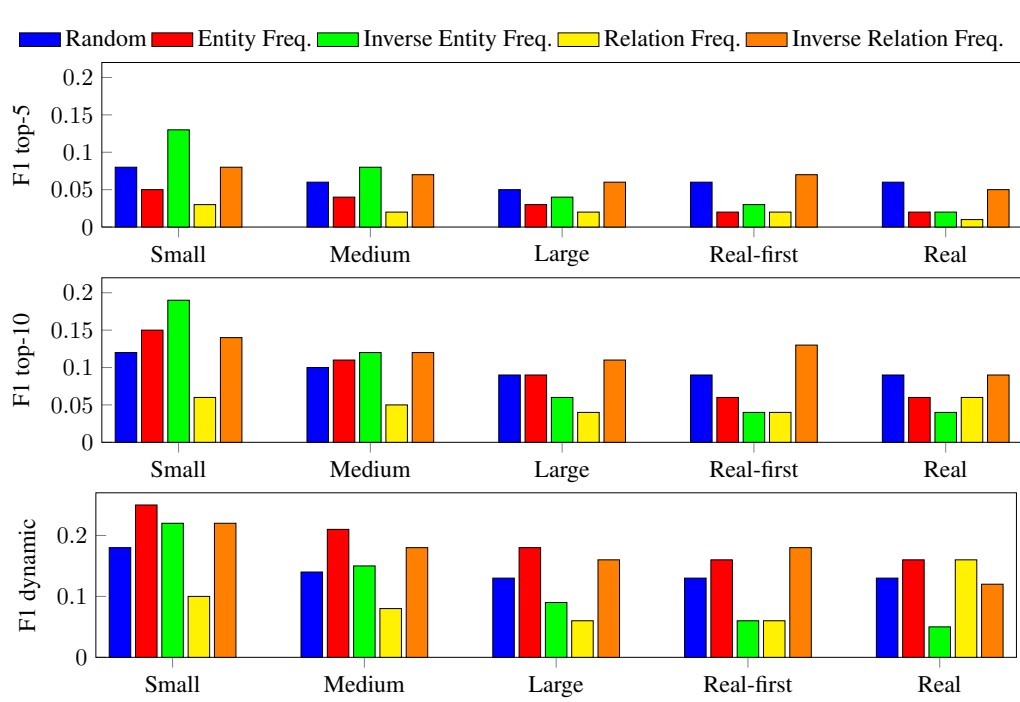

Figure 8: F1 for frequency statistics on WikiPro.

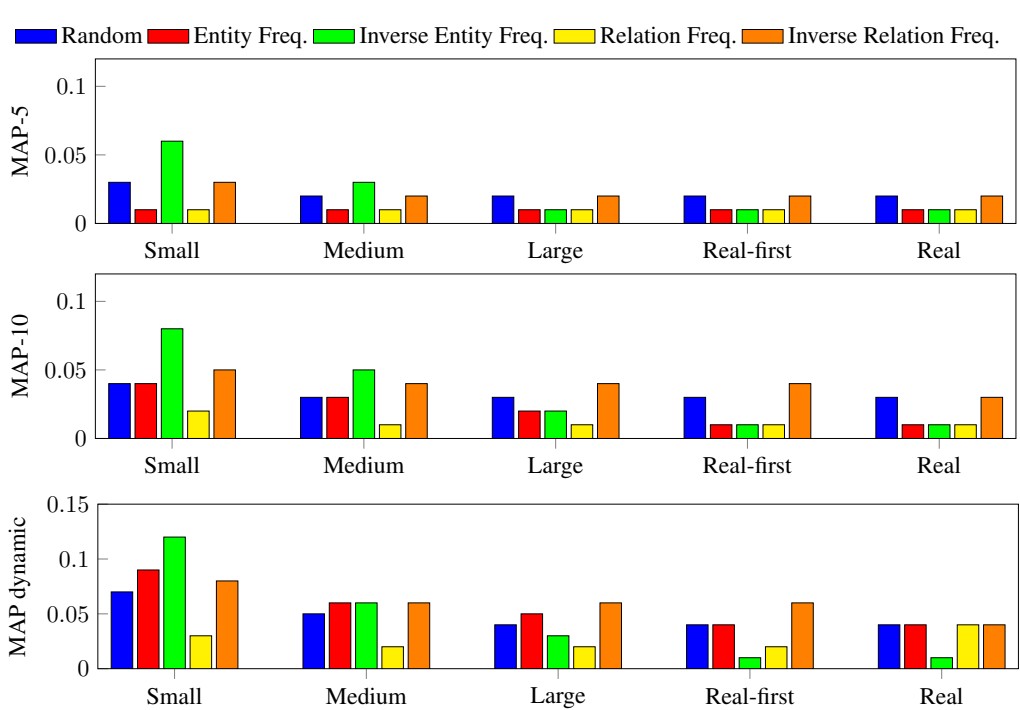

Figure 9: MAP for frequency statistics on WikiPro.

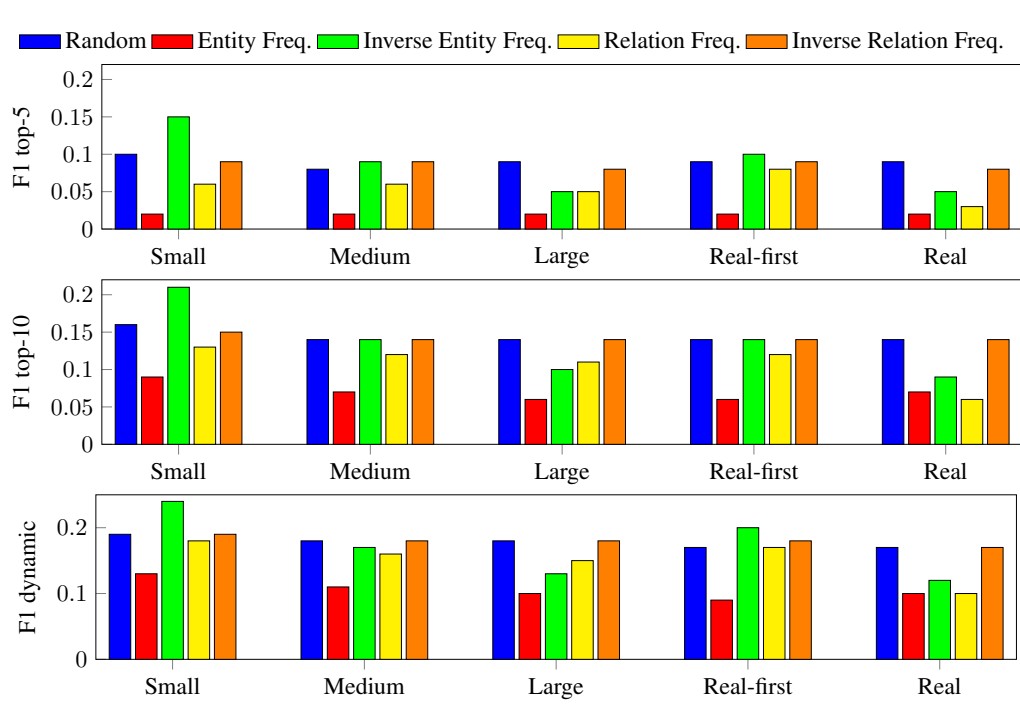

Figure 10: F1 for frequency statistics on WikiProFem.

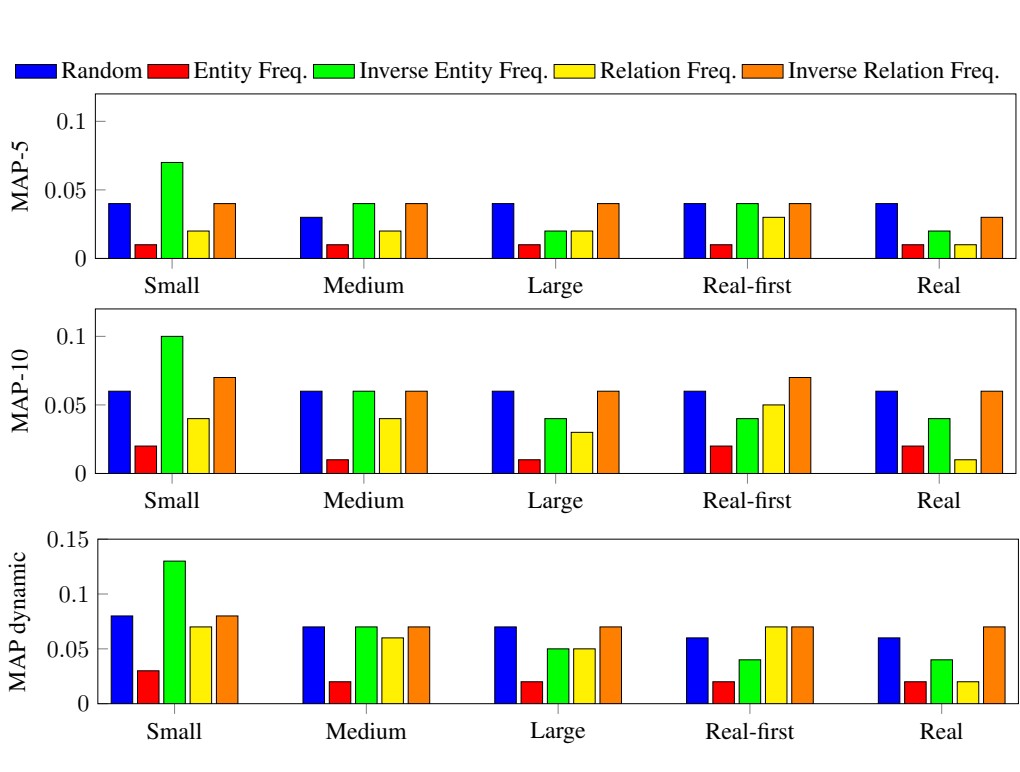

Figure 11: MAP for frequency statistics on WikiProFem.