# OpenReview forum: "Wiki Entity Summarization Benchmark"
_ICLR.cc/2025/Conference — ICLR 2025 Conference Withdrawn Submission_

### Official Review · Reviewer_xSZa · 2024-10-30

**Soundness:** 3
**Presentation:** 4
**Contribution:** 3
**Rating:** 6
**Confidence:** 5

**Summary:**

This paper presents a method to automatically create a dataset for evaluating entity summarization. Compared with existing datasets, with this new approach:
- larger datasets can be programmatically generated,
- mutli-hop (empirically 2-hop in the paper) relations are allowed,
- a connected subgraph is guaranteed to be sampled.

**Strengths:**

S1. A programmatically generatable benchmark for entity summarization. Can be larger and better than existing ones. It represents a useful contribution to the community. I like it.

S2. Generally reasonable approach to benchmark generation. The generated gold-standard summaries are likely to have high quality, despite lacking verification.

S3. Well-written paper. Very easy to read.

**Weaknesses:**

W1. Missing some important technical details and experiments related to the quality of the automatically generated gold-standard summaries.
- Line 175-176, how did you identify 'mentions' of Wikidata items in a Wikipedia page? If only relying on hyperlinks, recall could be low. Essentially it is an entity linking problem. I would like to see more details about its implementation. I also want to see a technical comparison of your approach with https://doi.org/10.1016/j.ipm.2013.12.001 which also relies on Wikipedia abstract to create gold-standard entity summaries.
- The above-mentioned entity linking, and your heuristic relation selection method, both can be inaccurate. Have you conducted experiments to evaluate the quality of your generated gold-standard summaries?
- Line 138-140, according to this problem definition, do you consider or ignore literals?

W2. While your sampled graphs can be arbitrarily large, it seems that your gold-standard summaries are limited in the following aspects.
- I am not sure whether generating a larger graph is really helpful for the entity summarization task. According to Table 1(b)(c)(d), while graphs differ in size, the number of gold-standard summaries remains almost unchanged, about 500 entities, if I understood correctly. You criticized existing benchmarks for their small size in terms of the number of entities with gold-standard summaries (e.g., 175 in ESBM), but your datasets are not significantly larger.
- All these ~500 entities are instances of person, while there are many other types of entities in DBpedia/Wikidata and in previous benchmarks like ESBM. Why did you limit your datasets to person entities? It introduces a bias.
- Only entities with English labels are considered. This is acceptable, but not necessary IMO.

**Questions:**

Apart from my questions in Weaknesses, I have the following further questions/comments.

Q1. For relation selection, you compared relation embedding with abstract embedding. The latter covers a very large piece of text. How about comparing with the embedding of the sentence where the relation value is mentioned?

Q2. Line 321-323, I did not understand why the bias comes from the small size and/or the incomplete edges of the sampled graphs. An extended explanation would be appreciated.

Q3. LinkSum performed best in your experiments. Is it possible that its good performance came from its use of Backlinks, which coincides with your approach to generating gold-standard summaries (based on entities mentioned in Wikipedia abstracts)? It may represent a bias of your ground truth.

---

> ### Author Response · Authors · 2024-11-22
> **Response (1/2)**
>
> > W1. Line 175-176, how did you identify 'mentions' of Wikidata items in a Wikipedia page? If only relying on hyperlinks, recall could be low.
>
> **Answer**: Our approach indeed identifies mentions of Wikidata items using Wikipedia hyperlinks in abstracts. We chose this method for its higher precision, as hyperlinks are manually curated, which is critical for generating high-quality ground-truth summaries for entity summarization. To address potential low mention counts on less popular pages, we provide two user-defined parameters: the initial set of input entities and the minimum number of hyperlinks in the abstract. This method is cost-efficient, precise, and easily accessible. The entire pipeline is publicly available, allowing researchers to customize components to suit their needs.
>
> In short, our entity mentions pipeline works as follows:
> 1. Extract a graph of relevant entities and relationships from Wikipedia hyperlinks and Wikidata.
> 2. Compute the intersection of these graphs, marking the triples as the summary.
> 3. Expand the Wikidata graph using a 2-hop random walk.
> 4. Add available node and label metadata.
>
> In cases with multiple edges between two entities, we use DistilBERT as a tiebreaker to select the most relevant label, choosing the one that best aligns with the context of the entire abstract.
>
> > technical comparison of your approach with https://doi.org/10.1016/j.ipm.2013.12.001 which also relies on Wikipedia abstract to create gold-standard entity summaries.
>
> **Answer**: There are key distinctions between WikES and the approach by Xu et al. [1]. First, we use Wikidata, while Xu et al. rely on DBpedia. Second, their method involves complex NLP techniques for entity linking and relation extraction, whereas our graph-based approach leverages the editorial work in Wikidata and Wikipedia. As mentioned earlier, entity linking improves recall but can reduce precision. Thus, Xu et al.'s approach is orthogonal to ours. We will add this duscussion and citation in the introduction of the revised paper.
>
> [1] Xu, D., Cheng, G., & Qu, Y. (2014). Preferences in Wikipedia abstracts: Empirical findings and implications for automatic entity summarization. Information Processing & Management, 50(2), 284-296.
>
> > W1 (2). Have you conducted experiments to evaluate the quality of your generated gold-standard summaries?
>
> **Answer**: We did consider evaluating the quality of the generated summaries. However, since Wikipedia itself serves as the ground truth and is curated by a large community of users, conducting a user study would not likely provide a better consensus than what Wikipedia already offers. To complement our analysis, we will include examples of the extracted summaries in the supplementary material.
>
>
> As an example, consider the ```Elvis Presley``` entity:
> ```
> (Elvis Presley)-[military unit]-> (32nd Cavalry Regiment)
> (Elvis Presley)-[genre]-> (blues)
> ...
> (Jim Morrison)-[influenced by]-> (Elvis Presley)
> (Elvis Country – I'm 10,000 Years Old)-[performer]-> (Elvis Presley)
> (The King)-[main subject]-> (Elvis Presley)
> ```
>
> The computed summary is
> ```
> (Elvis Presley)-[genre]-> (rockabilly)
> (Million Dollar Quartet)-[has part(s)]-> (Elvis Presley)
> (Jailhouse Rock)-[cast member]-> (Elvis Presley)
> ...
> (Viva Las Vegas)-[cast member]-> (Elvis Presley)
> (Elvis Presley)-[genre]-> (rhythm and blues)
> (Elvis Presley)-[record label]-> (Sun Records)
> (Elvis Presley)-[genre]-> (pop music)
> ```
>
> > W1 (3). Do you consider or ignore literals?
>
> **Answer**: We include literals as graph nodes during the random walk exploration.

---

> > ### Author Response · Authors · 2024-11-22
> > **Response (2/2)**
> >
> > > W2. While graphs differ in size, the number of gold-standard summaries remains almost unchanged.
> >
> > **Answer**: While our dataset generation pipeline can produce summaries with any number of entities, we demonstrate its capability with datasets containing 500 target entities, three times the size of ESBM. From these entities, we generated graphs as large as 230k nodes and 466k edges (WikiLitArt), which is two orders of magnitude larger than ESBM.
> >
> > > All these ~500 entities are instances of person.
> >
> > **Answer**: We selected target entities from a homogeneous group (instances of "person") to enforce semantic proximity, ensuring the graphs are both connected and relevant for the summarization task. As shown in Figure 2, this dataset avoids the frequency biases observed in ESBM. We acknowledge the importance of diversity and are committed to expanding the range of entity types in future work.
> >
> > > Only entities with English labels are considered.
> >
> > **Answer**:  We focused on the English-language version of Wikidata/Wikipedia because it is the largest and most comprehensive repository available. However, our methodology is language-agnostic and can be easily extended to support other languages.
> >
> > > Q1. How about comparing with the embedding of the sentence where the relation value is mentioned?
> >
> > **Answer**: Thank you for raising this interesting point. While enhancing our comparison method is worth exploring, our use of embeddings is limited to selecting the correct relationship in cases where two entities are connected by multiple edges. For this specific task, embeddings perform reasonably well, as illustrated in this [small set of examples](https://docs.google.com/spreadsheets/d/1J6C2yi5lv5RxbJsnAC274EfG0tP0p49FLtqe-qa3R3c/edit?usp=sharing), which compare the triple selected by DistilBERT ("picked") against all candidate triples.
> >
> >
> > > Q2. Line 321-323: why the bias comes from the small size and/or the incomplete edges of the sampled graphs.
> >
> > **Answer**: The bias arises from the limited size of the sampled graph, where only a small subset of nodes is selected. This subset may not accurately represent the label distribution or the overall structure of the larger graph, leading to biased results. The incomplete edges in the sampled graph further contribute to this bias, as they may not capture the full range of relationships present in the entire graph. We included this explanation in the revision.
> >
> > > Q3. LinkSum performed best in your experiments ... from its use of Backlinks
> >
> > **Answer**: The use of backlinks in LinkSum does resemble a random walk approach. However, while LinkSum performs best on WikiProFem, its performance varies across other datasets, showing alternating results. Therefore, while the backlink mechanism shares similarities with random walks, it cannot fully explain the performance observed in all cases. We included the explanation in the revised version.

---

> > > ### Comment · Reviewer_xSZa · 2024-11-22
> > >
> > > Thank you for your clarification. I think my score is already fairly high.

---

### Official Review · Reviewer_qY7Z · 2024-11-03

**Soundness:** 3
**Presentation:** 3
**Contribution:** 3
**Rating:** 6
**Confidence:** 4

**Summary:**

This paper introduced a novel large-scale benchmark, named as WIKES, for entity summarization (ES) in knowledge graphs. Existing benchmarks are limited in size and often rely on expensive and time-consuming manual annotations. In contrast, WIKES aims to overcome these limitations by using Wikipedia abstracts and knowledge graph structures from Wikidata to automatically generate high-quality entity summaries without the need for manual annotators.
This paper proposed a dataset generation algorithm by combining graph algorithms with natural language processing models. The automatic processing makes the dataset be scalable and suitable for multiple domains.
Finally, this paper  offers a comprehensive evaluation of several entity summarization methods, including both unsupervised (e.g., PageRank, RELIN, LinkSum) and supervised models (e.g., GATES). It highlights that unsupervised methods, particularly graph-based approaches, often outperform supervised models in large-scale knowledge graphs.

**Strengths:**

S1: The WIKES benchmark is scalable, leveraging automatic summary generation from Wikipedia and Wikidata without relying on costly manual annotations, making it applicable to large datasets across various domains. The use of random walk-based subgraph extraction ensures that the structure of knowledge graphs is preserved, capturing both topological and semantic complexities of entities while maintaining computational efficiency.

S2: This paper provides a thorough evaluation of multiple graph-based entity summarization methods (e.g., PageRank, RELIN, GATES), allowing for direct comparison of unsupervised and supervised approaches, highlighting the advantages of graph-based methods.

S3: The overall presentation is good and the authors provided source code for review.

**Weaknesses:**

W1: The summarization methods are limited to graph-based summarization techniques. The authors may need to evaluate some text generation methods. A broader comparison with recent NLP-based summarization techniques could add more depth.

W2: The paper focuses on scalability but only evaluating the small version of their dataset. The methods without efficiency concerns could be used to conduct evaluation on the large version to show the effectiveness of the proposed dataset.

W3: The random walk-based graph expansion approach may not always capture the most semantically relevant information for all types of entities. While the two-hop neighborhood approach is computationally efficient, it may miss out on key relationships that are further away in the graph but still contextually important for the entity. The authors could have considered a dynamic approach, where the hop count is adjustable based on the entity or relationship type.

W4: The authors could provide some qualitative examples of the generated summaries of model like LinkSum in Appendix .

**Questions:**

Q1: In your paper, entity summarization is derived from Wikipedia abstracts and infoboxes, primarily using extraction-based methods. However, you did not explore any text generation or abstractive summarization models (such as GPT or T5) to create summaries from the knowledge graph or the Wikipedia text. Given the recent advancements in text generation, have you considered evaluating the performance of generative models for entity summarization, especially in cases where the structured data might be sparse or incomplete? What are the potential reasons for not including these methods in your benchmark?

Typos:
- Line 34, Entity summarization (ES) => Entity Summarization (ES)
- Line 119 “(version 3.9) ? and”
- Line 122 “INFO” may need to be bold.

---

> ### Author Response · Authors · 2024-11-22
>
> > W1/Q1: The summarization methods are limited to graph-based summarization techniques ... explore text generation or abstractive summarization models (such as GPT or T5) to create summaries from the knowledge graph or the Wikipedia text.
>
> Thank you for the suggestion. While abstractive summarization models, such as GPT or T5, could serve as baselines, there are challenges in assessing their quality for our datasets. First, these models are trained on vast data, likely including Wikipedia, which introduces data leakage and makes evaluations unfair. Second, Large language models (LLMs) are prone to hallucinations, leading to inconsistencies in the evaluation pipeline compared to graph-based methods. Third, to our knowledge, there are no established methods for entity summarization using LLMs.
>
> > W2. The paper focuses on scalability but only evaluating the small version of their dataset.
>
> **Answer**: Thank you for raising this point. The results in Table 2 are limited to the smallest WikES dataset due to the inefficiency of previous methods on larger datasets. Yet, this smaller dataset enables meaningful comparisons in a controlled setting.
>
> To run experiments even on the smaller datasets, we made significant efforts to optimize and parallelize the implementation of prior methods. Adapting these methods to larger datasets, while technically feasible, would require substantial engineering effort beyond the scope of this work, which focuses on providing a robust pipeline for entity summarization.
>
>
> > W3: The authors could have considered a dynamic approach, where the hop count is adjustable based on the entity or relationship type.
>
> **Answer:** Thank you for the suggestion. While a dynamic hop count approach could treat each entity differently, it may also introduce frequency-related biases similar to ESBM and produce excessively large, uncontrolled graphs that do not represent the local neighborhood effectively. We opted for a simpler, theory-grounded solution that delivers the expected results efficiently, as shown in Figure 3.
>
>
> > W4: The authors could provide some qualitative examples of the generated summaries of model like LinkSum in Appendix.
>
> **Answer**: Below is a set of three examples of generated summaries using Linksum.
>
> Top-10 summary for ```Victor Hugo```:
>
> ```
> Victor Hugo, place of death, Paris
> Victor Hugo, place of burial, Panthéon
> Victor Hugo, child, Léopoldine Hugo
> Victor Hugo, spouse, Adèle Foucher
> Victor Hugo, father ->  Joseph Léopold Sigisbert Hugo
> Victor Hugo, father, Joseph Léopold Sigisbert Hugo
> Victor Hugo, mother, Sophie Trébuchet
> Victor Hugo, movement, French Romanticism
> Victor Hugo, notable work, Ninety-three(novel by Victor Victor Hugo, instance of,  human
> Victor Hugo, described by source, Library of the World's Best Literature
> ```
>
> Top-5 summary for ```Barbra Streisand```:
>
> ```
> Barbra Streisand, award received, Academy Award for Best Actress
> Barbra Streisand, record label, Columbia Records
> Barbra Streisand, influenced by, Billie Holiday
> Barbra Streisand, award received, Grammy Lifetime Achievement Award
> Barbra Streisand, award received, AFI Life Achievement Award
> ```
>
> Top 5 summary for ```Claude Debussy```:
>
> ```
> Claude Debussy, award received, Prix de Rome
> Claude Debussy, educated at, Conservatoire de Paris
> Pelléas et Mélisande, composer, Claude Debussy
> Claude Debussy, student of, Antoine François Marmontel
> Claude Debussy, place of birth, Saint-Germain-en-Laye
> ```

---

> > ### Comment · Reviewer_qY7Z · 2024-11-23
> > **To author**
> >
> > Thanks for your responses.
> >
> > Since most of my concerns have been addressed. I increase the points accordingly.

---

### Official Review · Reviewer_F6iB · 2024-11-03

**Soundness:** 2
**Presentation:** 2
**Contribution:** 3
**Rating:** 3
**Confidence:** 3

**Summary:**

This paper introduces a new entity summarization(ES) benchmark, called WIKES, sourced from Wikidata and Wikipedia. Compared existing ES benchmarks, WIKES is generated automatically without relying on human labeling. WIKES also contains complex topology and semantics of the knowledge graph by including 2-hop connected sub-graphs  and diverse topics.

**Strengths:**

1. The new benchmark WIKES is the first ES benchmark that does not require human annotation. And the generation method could be easily applied to generate other ES datasets with diverse topics and scales.
2. WIKES is the largest ES benchmark compared to existing benchmarks, which make it possible to explore the effectiveness of the ES methods over large scale datasets.
3. Some results on the smallest datasets of WIKES  are presented, giving baseline results for further researches.

**Weaknesses:**

1. Relying Wikipedia’s abstract to generate the ES datasets is cost-efficient and novel. But this makes the entity summarization generated based on the abstract  text rather than the triples of the entities in the knowledge graph. This might cause the entity summarization in WIKES not the gold entity summarization of the entities.
2. The DistillBERT is used to annotate the property that should be included in the summarization. The correctness of the final property is not evaluated, which is important to the quality of the WIKES in terms of entity summarization.
3. The datasets evaluation is not comprehensive. For example,
    (1). Figure 3 only shows the F1 evaluation on WIkiProFem, part of the WIKES benchmark. The F1 score on other subdatasets are not presented.
    (2). Table 2 shows that results of entity summarization methods on the smallest WIKES datasets. But the midium and the large WIKES datasets are not tested. It is not clear what would be the performance of current summarization methods on these two datasets.
4. The dataset quality are not analyzed, for example, the correctness and diversity that are important for ES benchmark.
5. Minor points and typos:
   (1). In line 119, there is an extra question mark after “(version 3.9)”.
   (2). The citation format in the main text seems not correct.

**Questions:**

1. Have you evaluated the correctness of property identification results based on the DistillBERT in terms of entity summarization?
2. Why the left side of the Equation (2) equal to the right side?
3. How would the minRW, maxRW, and minRW affect the random walk results for graphs in different scales? Especially how they would affect the quality of the entity summarization datasets?
4. What is the meaning of Real-first and Real in the Figure 3? They are not explained in the main text.

---

> ### Author Response · Authors · 2024-11-22
> **Response (part 1)**
>
> Thanks for the insightful questions. We have made an effort to clarify the points below.
>
> > Relying Wikipedia’s abstract to generate the ES datasets is cost-efficient and novel.
>
> **Answer**: Thanks for appreciating our core contributions.
>
> > W1. entity summarization generated based on the abstract text rather than the triples of the entities in the knowledge graph.
>
> **Answer**: Thank you for raising this point. While the Wikipedia abstract of an entity does semantically represent its summary, it also serves as the counterpart to a Wikidata entity. By carefully mapping each entity and its relationships to the corresponding Wikipedia page, we are able to extract the most representative information that Wikipedia contributors have synthesized in the abstract, ensuring alignment with the knowledge graph.
>
> > W2. The DistillBERT correctness of the final property is not evaluated
>
> Evaluating DistilBERT's predictive correctness is challenging without ground truth. Yet, we emphasize that its role in our pipeline is limited to selecting a triple when multiple relationships exist between two nodes—a scenario that arises in only 20% of edges and rarely involves more than two triples.
>
> We chose DistilBERT, a robust and scalable model, over manual annotation for this task. For example, given the abstract:
>
> ```The album featured 'Blowin' in the Wind' and 'A Hard Rain's a-Gonna Fall,' which adapted the tunes and phrasing of older folk songs,```
>
> and the corresponding triples in Wikidata
>
> ```
> (Bob Dylan)-[notable work]->(Blowin' in the Wind)
> (Blowin' in the Wind)-[composer]->(Bob Dylan)
> (Blowin' in the Wind)-[lyrics by]->(Bob Dylan)
> (Blowin' in the Wind)-[performer]->(Bob Dylan)
> ```
>
> DistilBERT selected the triple ```(Blowin' in the Wind)-[lyrics by]->(Bob Dylan)``` from several valid options, a choice we found reasonable but hard to verify.
>
> To provide transparency, we included a [set of examples](https://docs.google.com/spreadsheets/d/1J6C2yi5lv5RxbJsnAC274EfG0tP0p49FLtqe-qa3R3c/edit?usp=sharing) comparing the triple selected by DistilBERT (column "picked") against all candidate triples between two node, demonstrating the model's general effectiveness in selecting meaningful triples.
>
> In this case, a human annotator could reasonably select any of these triples. DistilBERT chose the third triple, (Blowin' in the Wind)-[lyrics by]->(Bob Dylan), which we consider a reasonable choice. Based on manual inspection of several examples, we found DistilBERT's selections to be generally appropriate.
>
> To provide transparency, we have created a [small set of examples](https://docs.google.com/spreadsheets/d/1J6C2yi5lv5RxbJsnAC274EfG0tP0p49FLtqe-qa3R3c/edit?usp=sharing), comparing the triple selected by DistilBERT (column "picked") against all candidate triples between two nodes. These examples demonstrate the model’s reasonable performance in selecting meaningful triples.
>
> > W3(1). Figure 3 only shows the F1 evaluation on WIkiProFem
>
> **Answer**: Thank you for highlighting this omission. Due to space constraints, we included results for only one representative dataset, WikiProFem. However, we conducted experiments on all datasets, and the findings are consistent across them. The detailed results are now included in the supplementary material (Figures 4–11) for completeness.
>
> > W3(2). Table 2 shows that results of entity summarization methods on the smallest WIKES datasets.
>
> **Answer**: Thank you for raising this point. The results in Table 2 are limited to the smallest WikES dataset due to the inefficiency of previous methods on larger datasets. Yet, this smaller dataset enables meaningful comparisons in a controlled setting.
>
> To run experiments even on the smaller datasets, we made significant efforts to optimize and parallelize the implementation of prior methods. Adapting these methods to larger datasets, while technically feasible, would require substantial engineering effort beyond the scope of this work, which focuses on providing a robust pipeline for entity summarization.

---

> > ### Author Response · Authors · 2024-11-22
> > **Response (Part 2)**
> >
> > > Minor points and typos.
> >
> > **Answer**: Thanks. We fixed them.
> >
> > > Q1. Have you evaluated the correctness of property identification results based on the DistillBERT in terms of entity summarization?
> >
> > **Answer**: See our answer to W2.
> >
> > > Q2. Why the left side of the Equation (2) equal to the right side?
> >
> > **Answer**: The left side of Equation (2) is equivalent to the right side under the Markov property, which states that the probability of transitioning to a new state depends solely on the previous state.
> >
> > > Q3. How would the minRW, maxRW, and minRW affect the random walk results for graphs in different scales?
> >
> > **Answer**: The parameters **minRW** and **maxRW** define the number of walks and the extent of node exploration. Larger values explore more nodes, generating broader subgraphs suitable for large graphs, while smaller values focus on nearby nodes, creating more localized subgraphs. This allows the random walk to adapt to the graph's size and complexity effectively.
> >
> >
> > > Q4. meaning of Real-first and Real in the Figure 3?
> >
> > **Answer**: We apologize for the omission. **Real-first** refers to frequencies of nodes and relationships calculated within the first-hop neighborhood of each target node in the original Wikidata knowledge graph. **Real** represents the frequencies calculated from the entire Wikidata knowledge graph.

---

> > > ### Author Response · Authors · 2024-11-25
> > >
> > > Thanks again for the insightful feedback and appreciating some chief parts of our methodology. If there is any other clarification you would like to pose, we are happy to answer.

---

### Official Review · Reviewer_mGWX · 2024-11-11

**Soundness:** 2
**Presentation:** 2
**Contribution:** 2
**Rating:** 3
**Confidence:** 4

**Summary:**

The paper introduces a benchmark dataset for Wiki entity summarization called WikES with the help of random walks.

**Strengths:**

--> The authors introduce a benchmark dataset for entity summarization over wikidata.

**Weaknesses:**

--> Why the automated mapping from Wikipedia to Wikidata allows for better summarization over entities as stated in "Wikidata to automatically map entities from Wikipedia to Wikidata. This automation allows us to efficiently generate summaries for any number of entities"
--> The motivation behind choosing these 4 algorithms for comparison is missing.
--> A thorough comparison with all the benchmark datasets given in the related work is missing.
--> The main contribution on why this benchmark dataset is needed should be motivated more clearly.
--> The overall results in table two are very low, what would be the reason behind that?

**Questions:**

--> Why the automated mapping from Wikipedia to Wikidata allows for better summarization over entities as stated in "Wikidata to automatically map entities from Wikipedia to Wikidata. This automation allows us to efficiently generate summaries for any number of entities"
--> Why did authors only choose these algorithms for comparison?
--> Why these algorithms are not also tested on all the benchmark datasets which are given in the related work?

---

> ### Author Response · Authors · 2024-11-22
>
> We thank the reviewer for the comments. We acknowledge that most of the confusion is on the term _benchmark_. Our main contribution is on a pipeline for realistic dataset generation for entity summarization.
>
> > W1. Why the automated mapping from Wikipedia to Wikidata allows for better summarization over entities
>
> **Answer:** We refer to the efficiency of generating summaries through automated mapping from Wikipedia to Wikidata, not the quality. Unlike manual annotation, which is costly and time-intensive, our approach scales effortlessly to thousands of entities with minimal cost, requiring only computational resources. We have clarified this in the revised manuscript.
>
> > W2.The motivation behind choosing these 4 algorithms for comparison is missing.
>
>
> **Answer:** We selected top-performing models on ESBM, excluding those that failed to return results within two days or relied on unavailable external knowledge. For example, BAFREC [1] could not generate results on the small graph (13000) within two days, and MPSUM [2] did not terminate even after 15 days. INFO [3] was excluded because it heavily depends on external knowledge that is no longer accessible.
>
> This ensures our evaluation is both practical and reproducible under reasonable computational constraints.
>
>
> [1] Hermann Kroll et al. Bafrec: Balancing frequency and rarity for entity characterization in
> linked open data. 1st International Workshop on Entity REtrieval (EntRE), 2018.
>
> [2] Dongjun Wei et al. Mpsum: entity summarization with predicate-based matching. EYRE@CIKM’2018, 2018.
>
> [3] Gong Cheng et al. Generating characteristic summaries for entity descriptions. TKDE, 35(5): 4825–4835, 2023.
>
> > W3. A thorough comparison with all the benchmark datasets given in the related work is missing
>
> **Answer:** We acknowledge that the term "benchmark" may have been overloaded in our discussion. Our primary contribution lies in the automatic generation of datasets for entity summarization, not in providing a comprehensive benchmarking framework. A full benchmarking analysis is left for future work. For comparisons with the existing ESBM benchmark, we refer to https://github.com/nju-websoft/ESBM/tree/master/v1.2. We have clarified our contributions in the revised manuscript.
>
> > W4. The main contribution on why this benchmark dataset is needed should be motivated more clearly.
>
> **Answer:** We acknowledge that the motivation for our benchmark could have been more clearly emphasized. The only existing benchmark, ESBM—of which INFO is a part—includes just 175 manually annotated entities. ESBM has significant biases (see Figure 2) toward high-degree entities and relationship frequency, which allows a simple baseline that selects triples with the most frequent relationships to outperform some existing algorithms.
>
> In contrast, WikES is a large, automatically generated, and realistic _pipeline for generating datasets_ that accurately reflects the characteristics of the underlying Wikidata knowledge graph. As we showcase on four datasets in Figure 3, the WikES pipeline generates more representative and unbiased datasets, addressing the limitations of existing benchmarks and enabling more robust evaluation of entity summarization methods.
>
> > W5. The overall results in table two are very low, what would be the reason behind that?
>
> **Answer:** Thank you for this observation. The low overall results reflect how previous algorithms are tailored to outperform ESBM, effectively overfitting to its specific characteristics. As shown in Figure 2, ESBM is biased toward entity and relationship frequency, and many of these algorithms exhibit similar biases. In contrast, WikES is built on different principles, revealing the limitations of these approaches and underscoring the need for further research in this critical area.

---

> > ### Author Response · Authors · 2024-11-25
> >
> > Thanks again for the insightful feedback. We are committed to provide extra details whether needed. On the other hand, if you find the responses satisfactory please consider raising the score.

---

### Author Response · Authors · 2024-11-22
**Confusion on term benchmark**

We thank the reviewers for the feedback on our submission. We acknowledge that the use of the term _benchmark_ might be misleading. Our main contribution is a **pipeline for generating datasets** rather than a complete benchmarking of the state-of-the-art. We believe this kind of contribution is of interest for the research community and show insights on why our dataset deviates from the standard ESBM benchmark.

---

### Note · Authors · 2025-04-28

I have read and agree with the venue's withdrawal policy on behalf of myself and my co-authors.

---

### Meta-Review · Area_Chair_pqtq · 2024-12-18

**Metareview:**

This paper introduces Wiki Entity Summarization Benchmark (WikES) which is really an approach to generate entity summarization datasets. While reviewers agreed that this constitutes a potentially useful direction, the motivation and even the characterization of WikES as a "benchmark" seems somewhat confusing. For example, as pointed out by reviewer F6iB, there is limited evaluation of the generated data, though this seems critical for a benchmarking paper.

**Additional Comments On Reviewer Discussion:**

The authors addressed some of the important points raised by reviewers, or at least offered clarifications. It is unfortunate that F6iB and mGWX did not engage in the discussion period, but some of the issues raised by the latter, especially, seem insufficiently addressed in rebuttal.

---

### Decision · Program_Chairs · 2025-01-22

Reject